# Fluid face but not gender: Enfacement illusion through digital face filters does not affect gender identity

**Luca Provenzano**[1]☯*, **Hanna Gohlke**[2]☯, **Gianluca Saetta**[2,4], **Ilaria Bufalari**[3], **Bigna Lenggenhager**[2]‡, **Marte Roel Lesur**[2,5]‡

**1** Center for Life Nano- & Neuro-Science, Italian Institute of Technology, Rome, Italy, **2** Department of Psychology, University of Zurich, Zurich, Switzerland, **3** Department of Psychology of Developmental and Socialization Processes, Sapienza University of Rome, Rome, Italy, **4** Professorship for Social Brain Sciences, Department of Humanities, Social and Political Sciences, ETH Zurich, Zurich, Switzerland, **5** Department of Computer Science and Engineering, Universidad Carlos III de Madrid, Madrid, Spain

☯ These authors contributed equally to this work.
‡ BL and MRL also contributed equally to this work.
* luca.provenzano@iit.it, luca.provenzano.papi@gmail.com

## Abstract

It has been shown that observing a face being touched or moving in synchrony with our own face increases self-identification with the former which might alter both cognitive and affective processes. The induction of this phenomenon, termed enfacement illusion, has often relied on laboratory tools that are unavailable to a large audience. However, digital face filters applications are nowadays regularly used and might provide an interesting tool to study similar mechanisms in a wider population. Digital filters are able to render our faces in real time while changing important facial features, for example, rendering them more masculine or feminine according to normative standards. Recent literature using full-body illusions has shown that participants' own gender identity shifts when embodying a different gendered avatar. Here we studied whether participants' filtered faces, observed while moving in synchrony with their own face, may induce an enfacement illusion and if so, modulate their gender identity. We collected data from 35 female and 33 male participants who observed a stereotypically gender mismatched version of themselves either moving synchronously or asynchronously with their own face on a screen. Our findings showed a successful induction of the enfacement illusion in the synchronous condition according to a questionnaire addressing the feelings of ownership, agency and perceived similarity. However, we found no evidence of gender identity being modulated, neither in explicit nor in implicit measures of gender identification. We discuss the distinction between full-body and facial processing and the relevance of studying widely accessible devices that may impact the sense of a bodily self and our cognition, emotion and behaviour.

## Introduction

How we sense our body is intimately linked to our self-identity. This bodily self-identity is the result of the incessant integration of multiple bodily signals from both inside and outside our

**Data Availability Statement:** All Fluid face but not gender: enfacement illusion through digital face filters does not affect gender identity files are

available from the OSF database (https://osf.io/bpa2u/).

**Funding:** B. L. and M. R. L. were funded by the Swiss National Science Foundation (SNSF; PP00P1_170511, PP00P1_202674) https://www.snf.ch/en The funders had no role in study design, data collection and analysis, decision to publish, or preparation of the manuscript.

**Competing interests:** The authors have declared that no competing interests exist.

body [1]. Moreover, this constant stream of bodily signals also affects cognitive and affective processes, i.e., embodied cognition, [2], such as emotion perception [3] and decision making [4, 5]. Experimental manipulations involving conflicting signals regarding one's own body to alter the sense of body ownership have shown to also affect higher-level self-concept, cognition, emotion and behaviour [6–8]. In particular, the embodiment illusion [9–11] uses temporarily and spatially congruent sensory information between one's own body and another seen body to alter bodily self- identity.

Recently, several studies have used such multisensory stimulation paradigms to investigate how embodying a different gender might alter one's own self-concept [12], and in particular gender identity [13]. Tacikowski and co-authors [13], showed that by providing synchronous visuotactile cues to a gender-mismatched full body observed from a first-person view (egocentric point of reference), the participants' own gender identity shifted towards a more balanced identification with both genders according to explicit and implicit measures. Remarkably, this shift in gender identity was associated with the degree of embodiment experienced over the seen body, i.e., how strongly participants reported that the observed body was their own.

In this study we wanted to replicate the findings by Tacikowski and colleagues [13] by using the enfacement illusion instead of a full body illusion. The enfacement illusion refers to an embodiment paradigm in which a combination of sensory cues such as touch [14–16], or movement/proprioception [17–19], is presented congruently between one's own face and the allocentrically observed face of another person, like in a mirror reflection. When the bodily signals experienced on one's own face, i.e., movement and touch, are temporally and spatially congruent to those observed on the seen face, participants may perceive another person's face more as their own [14, 19] compared to when movement or touch are temporally and spatially incongruent. As our face possesses unique physical features compared to the rest of the body that distinguish us socially and give emotional cues to others [16], with face perception having a key role in cognition since our first days of life [20], the enfacement paradigm has been extensively used to study the plasticity of facial self-identity [14, 15] and how this might in turn alter (social) cognition [21].

While in the past the enfacement illusion has been exclusively an experimental manipulation confined to laboratory settings, the recent technical development and availability of smartphones and digital face filters have provided the general population with the tools to, for example, render their faces more beautiful according to normative standards by enhancing certain more socially desired facial features, or change fundamental aspects of their facial identity such as age or gender [22]. Moreover, applications with massive numbers of daily users like Instagram and Snapchat have massively increased our confrontation with real time rendered reflections of ourselves [23] and self-portraying behaviours [24]. With the ubiquity of such digital filters in everyday life, a new type of body image distortion named "snapchat dysmorphia" has been recently described [25]. This term refers to the increasing observation by cosmetic surgeons of more patients wanting to undergo plastic surgery to look like the filtered versions of themselves [26].

Given the growing use of these tools and the increasing prevalence of related body image distortion conditions, it deems important to study how such filters might alter our sense of body and self-identity. Therefore, in the current study we investigated how filters that manipulate stereotypical cues associated with gender (My Twin, Snap inc.) might alter gender identity. As previously stated, the study not only aims to extrapolate the findings by Tacikowski and co-authors [13] to a face rather than a full-body setting, but also to assess the effects of an arguably more ecologically relevant intervention (i.e., widely used face filters in contrast to a costly laboratory-based method). For instance, despite the setup used by Tacikowski and colleagues being arguably more complex, recent findings collected from male participants enfacing a

female face showed a reduction of implicit associations between gender and scientific disciplines [17]. In the current study, participants' appearance was altered to that of a different gender according to stereotypical visual cues (see Fig 2). We further manipulated the temporal synchrony between the own and the seen face movements, which is regularly used as a control condition in the enfacement and embodiment literature (e.g., [8, 14]). We measured the degree of self-identification and agency of the seen face through a short enfacement questionnaire [18] that additionally included items regarding similarity, attractiveness and emotional response to the seen face [15]. Furthermore, we assessed participants' gender identity with an explicit questionnaire, the Revised BEM Sex-Role Inventory [27], as well as an auditory version of the brief gender implicit association test (IAT) [13]. We expected that participants would self-identify more with the seen face and experience a stronger sense of agency in the synchronous compared to the asynchronous condition. In line with previous studies using the enfacement illusion, synchronous stimulation was further expected to enhance perceived similarity [8] and attractiveness [28] of the observed face. Finally, we expected that participants' gender identity would shift more towards the opposite of the self-identified gender in the synchronous compared to the asynchronous condition.

## Methods and materials

### Participants

Based on the study of Tacikowski and colleagues [13], we recruited a total of 69 right-handed participants. Data from one male participant were excluded due to the refusal of sharing anonymous data on open platforms, resulting in a total sample size of 68 participants (33 males (Age mean = 25.18, SD = 6.56, range: [19, 44]) and 35 females (Age mean = 22.06, SD = 3.56, range: [18, 36]). Two participants were left-handed. Healthy subjects between 18 and 45 years of age and without a history of neurological or psychiatric disorders were recruited through a mailing list and an online platform at the Zurich University. Participants received a compensation of one course credit or 15 CHF for participating in the study. No participant identified themselves as non-binary.

All participants described themselves as cisgender, i.e., their internal sense of gender corresponded to their biological sex identified at birth. Participants were recruited between March and July 2021. All data was collected and held anonymously.

### Experimental apparatus

The experimental tasks were developed using Unity (version 2018.2; Unity Technologies, San Francisco California) to display the tasks and render the participants' image on the computer display. Participants' faces were captured by using a 1280x720 resolution webcam and were altered in real time by the Snap Camera software (Snap Inc., Santa Monica, CA) running the My Twin (https://lens.snapchat.com/a1cef9ac85c04cd0995858d770754eac) and My Other Twin (https://lens.snapchat.com/c24795ed958b4dda950efdffec0a490e) digital filters. Data recording using these filters was performed in the period ranging from 09.03.2021 to 12.05.2021. The video feed from the Snap Camera software was projected into a mesh corresponding to the aspect ratio of the webcam using Unity.

### Experimental procedure

Upon arrival, participants were asked to read and sign the study written informed consent and a COVID-19 screening form. They were asked to report their age, indicate their dominant hand and to identify their gender as either masculine, feminine or non-binary. They were

never asked to report their biological sex and their self-identified gender was considered relevant for the rest of the procedure. Due to COVID-19 restrictions prohibiting two people in the same room without protective masks, participants were informed that they would be monitored through an additional computer with a webcam that streamed the whole procedure to another room, where the researcher observed the experiment. All the experimental tasks were performed while sitting in front of a computer.

The overview of the experimental procedure is presented in Fig 1.

First, participants completed the Revised BEM Sex-Role Inventory [29] to measure individuals' femininity and masculinity and rated on a Visual Analogue Scale (VAS) how feminine and masculine they felt at baseline, i.e., before experiencing any experimental manipulation (please refer to the Measures paragraph for a detailed description of these measures). After that, in the enfacement phase, participants saw the filtered version of their face (see Fig 2) in either a synchronous or asynchronous condition. In the synchronous condition participants observed the filtered face moving synchronously with their own facial movements, whereas in the asynchronous condition the video feedback was delayed by 1s. Both the synchronous and the asynchronous stimulation lasted 60 seconds. Immediately after each condition, participants completed an auditory version of the brief gender IAT [13] while still looking at their filtered face (delayed or not according to the condition). Upon completion, participants were asked to answer both the BSRI and the masculinity/femininity VAS. Lastly, we measured the strength of the enfacement illusion through an enfacement questionnaire and collected the ratings for similarity and attractiveness of the observed filtered face as well as the participant's emotional state. This procedure, from the enfacement phase to the post-enfacement measures, was repeated in a counterbalanced order, so that each participant experienced both conditions (synchronous and asynchronous). The whole experiment lasted approximately 30 minutes.

## Enfacement phase

Before the enfacement procedure, participants were presented with written instruction that invited them to keep their hands on their knees and to explore the new filtered version of their

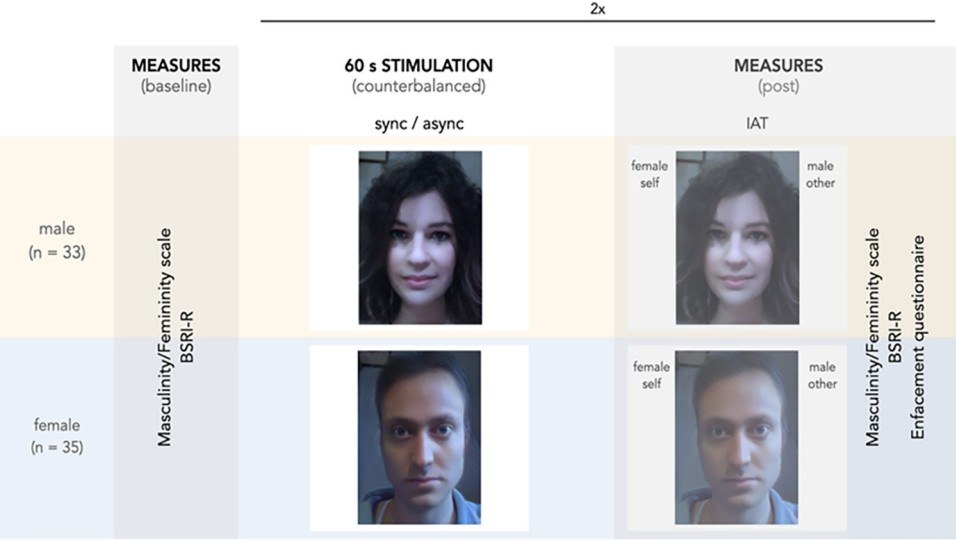

**Fig 1. Overview of experimental procedure including the baseline measures, the enfacement phase, IAT and the post-manipulation measurements.** Printed under a CC BY license, with permission from Gohlke Hanna and Drimmer Maxwell, original copyright 2023.

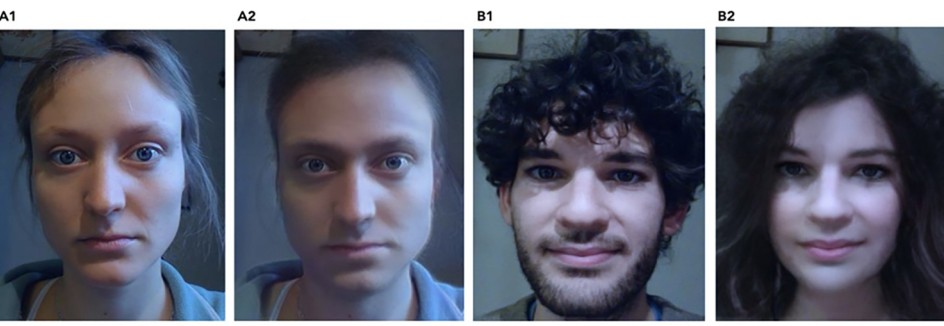

**Fig 2. Depictions of My Twin filter.** (A1) Live capture of a woman's face without any filters applied. (A2) Live capture of a woman's face with a male filter applied. This male filter adds a paler lip colour, removes long hair, and enhances the thicknesses of eyebrows. Furthermore, it changes the shape of the eyes and lips, enhances the squareness of the jaw, and adds facial hair. (B1) Live capture of man's face without any filters applied. (B2) Live capture of a man's face with the female filter applied. This female filter adds a pinkish lip colour, eye make-up, middle length hair, and makes the eyebrows thinner. Furthermore, it changes the shape of the eyes and lips, lightens and smoothens the skin, and enhances the angularity of the jaw while removing any facial hair. Printed under a CC BY license, with permission from Gohlke Hanna and Drimmer Maxwell, original copyright 2023.

face, which was presented on the screen in a mirror-like fashion. They were asked to perform natural movements such as blinking, opening and closing their mouth, moving their cheeks, and tilting their head in all directions. Participants were informed that they were about to see their own faces with an overlay filter. The temporal synchrony of the observed filtered face was manipulated within subjects. In the synchronous condition participants perceived their filtered face as if they were looking into a mirror, with their movements being synchronously replicated by the filtered face they were observing on the computer monitor. In the asynchronous condition there was a 1s delay between participants' facial movements and the movements observed on their filtered face. The gender of the filter was always the mismatching gender, i.e., self-identified female participants experienced a stereotypically masculine version of themselves, whereas self-identified male participants experienced a stereotypically feminine version (Fig 2). The synchronous and asynchronous enfacement condition lasted 60 seconds each.

## Measures

**Revised BEM Sex-Role Inventory (BSRI-R).** The original BEM Sex-Role Inventory (BSRI) [29] describes feminine and masculine stereotypes based on social desirability in the 1970s [29]. Whereas masculinity refers to instrumental and acting aspects, such as being assertive, acting like a leader etc., femininity is assessed using relationship oriented and expressive items, i.e., being cooperative, caring and dependent [27, 30]. Over the last decades, shifts in gender roles have changed social expectations towards men and women [27]. Troche and Rammsayer [27] developed a revised version of the German BEM Sex-Role Inventory (BSRI-R) [31]. The BSRI-R consists of 15 items each for the male and female scale, capturing two independent factors [27]. For this study, ten items for the male (i.e., ambitious, dominant, have leadership skills, willing to take a stand, competitive, assertive, a strong personality, forceful, act like a leader, aggressive) and female (affectionate, sensitive to other's needs, tender, sympathetic, loyal, eager to soothe hurt feeling, understanding, gentle, compassionate, warm) scale were chosen based on the item-total correlations. For simplicity we call these congruent and incongruent items. Participants were asked to rate how much each of the statements applied to themselves using a 7-point Likert scale ranging from "not at all" (0) to "very much" (7).

**Masculinity/Femininity Visual Analogue Scale (VAS).** As an explicit gender identity measure, we asked participants to rate how masculine/feminine they felt in that particular moment on a visual analogue scale (VAS) ranging from "very masculine/feminine" to "neither masculine nor feminine" to "very feminine/masculine" [13]. This scale did not contain any numbers but was coded respectively from 0–1. Scale assignment was different for male and female participants, such that the self-identified gender at the beginning of the experiment was presented on the right-side end of the scale (respectively coded as 1).

**Auditory gender identity Implicit Association Test (IAT).** To assess implicit gender identity, we used the auditory version of the brief gender identity IAT [13, 32]. Participants listened to 20 words that belonged to one of four different categories (male, female, self, and other). The stimuli consisted of twenty words that were read by a German native speaker. The volume of each word was normalised and edited to have equal length (0.8s) using Audacity 2.4.2 (https://audacityteam.org). The ten words belonging to the categories "self" (i.e., I, self, me, my, mine) and "other" (they, them, others, their, they) were translated to German from the gender identity IAT [32]. All female (i.e., Julia, Anna, Laura, Michaela, Sofie) and male (e.g., Lukas, Daniel, Paul, Thomas, Johannes) names correspond to the German version of the gender career IAT [33].

Participants were instructed to classify each word into one of the four categories ("female", "male", "self" and "other") which were visually presented on the left and right side of the screen while they were still seeing their filtered face shown in synchrony or asynchrony (depending on condition) in the centre of the screen (see Fig 1), by pressing either the left or the right mouse button. In the congruent IAT condition the participants' gender corresponded to the gender associated with themselves. For example, in a congruent IAT block a female participant would see "self" and "female" presented together on one side of the screen and respond with the same corresponding mouse button to words belonging to these two categories. In contrast, in the incongruent IAT condition, the participants' gender did not correspond to the gender associated with self. In other words, in an incongruent IAT block a female participant saw "self" and "male" presented together on one side of the screen and responded with the corresponding mouse button to words belonging to these. Each IAT block consisted of 60 stimuli (three repetitions of all 20 items) presented in random order. After each a/synchronous stimulation condition participants performed both the congruent and incongruent IAT blocks (order of presentation was randomised). Participants had a maximum of 3 s to react to each stimulus. If no key was pressed within this period or if participants gave a wrong answer, they heard a "wrong" feedback sound.

**Enfacement questionnaire.** To measure the strength of the enfacement illusion, participants answered a shortened version of the Enfacement Questionnaire [18] at the end of each experimental block. The questionnaire included one question for the ownership and one for the agency component of the enfacement illusion, as well as one control question. Additional ratings for the attractiveness and similarity of the observed faces were collected. Finally, we included one last item measuring the emotional state after the enfacement (see Table 1).

**Table 1. Enfacement questionnaire items.**

| Enfacement Questionnaire | Subscale |
| --- | --- |
| I had the impression to look at myself in a mirror | Ownership |
| It seemed as if I was in control of the person's face movement | Agency |
| It seemed as if I might have more than one face | Control Item |
| How similar was the seen face to your own? | Similarity |
| How attractive was the face you were seeing? | Attractiveness |
| During the last session I felt: | Emotional State |

For the items measuring the enfacement illusion components, participants had to answer using a VAS ranging from "strongly disagree" to "neither agree nor disagree" to "strongly agree". For the ratings of similarity, we used a VAS ranging from "very different" to "neither similar nor different" to "very similar". For the ratings of attractiveness, we used a VAS ranging from "very unattractive" to "neither attractive nor unattractive" to "very attractive". Finally, the VAS used to assess the emotional state ranged from "very negative" to "neither positive nor negative" to "very positive". All the scales were respectively coded with values between 0 and 1 (with the neutral label corresponding to 0.5), which were not presented to the participants.

## Data analysis and data processing

R studio v. 1.3 was used for statistical analyses. Data and R script used for analysis are publicly available on OSF (https://osf.io/msvbz), together with the pre-registered hypotheses. Below we report additional analysis for consistency between our measures and other studies, however all additional pre-registered analysis can be found on the Supplementary Materials. Data were tested for normality through statistical testing (Shapiro-Wilk-Normality-Test) and visual inspection of the residuals of histograms and QQ-plots. For each measure, we calculated an ANOVA considering the between-factor *Gender* (male/female), the within-factor *Synchrony* (synchronous/asynchronous) and their interaction. The tested model across all the analyses was the following:

$$DV = intercept + \beta1\ (Gender) + \beta2\ (Synchrony) + \beta3\ (Gender\ by\ Synchrony) + e$$

where "$\beta_x$" stands for the estimated parameters, "$e$" stands for the residuals. If the residuals did not fulfil the prerequisite of normality and/or homogeneity, an aligned ranks transformation analysis of variance (ANOVA) for nonparametric factorial analyses (ART–ANOVA) [34] was performed.

The findings were visualised using pirate plots, which were generated in R through the *yaRrr* package [35]. In this plot, the points represent raw data (jittered horizontally), the vertical bar and the inference band show the mean and the 95% Confidence Intervals (CIs), and the bean displays the data density. To assess changes from baseline for the relevant measures, we applied a 1-sided t-test.

For the IAT, participants on average classified the items correctly in 93.46% (SD = 8.68%) of all trials. Three participants had to be excluded from the analysis due to the low accuracy rate (<80%) in the task, resulting in a sample of sixty-five participants (35 females and 30 males). As in Tacikowski and co-authors [13] all trials with a reaction time shorter than 200ms and longer than 1500ms were furthermore excluded. After this exclusion on average 85.14% (SD: 13.51) of the correct trials per participant were included for further analysis. The final sample consisted of 12,622 reaction times, which is 77,76% of the original data of the 68 participants. Reaction times were log-transformed [13].

## Ethics statement

The study has been approved by the Ethics Committee of the Faculty of Arts and Social Sciences at Zurich University (Approval Number 17.12.15 for "Behavioural studies investigating development and plasticity of the bodily self") and was conducted at the Department of Psychology at Zurich University. The individuals pictured in Figs 1 and 2 have provided written informed consent (as outlined in PLOS consent form) to publish their image alongside the manuscript.

## Results

### Strength of the enfacement illusion

**Sense of ownership.** The ART-ANOVA results showed a significant main effect of *Synchrony* [$F(1,66) = 7.99$, p = 0.006, $\eta_p^2 = 0.108$); Fig 3], no main effect of *Gender* (p = 0.94, $\eta_p^2 = 0.0001$) and no significant interaction of *Gender* by *Synchrony* (p = 0.95, $\eta_p^2 < 0.0001$). As we expected, participants experienced a significantly stronger enfacement illusion during the synchronous compared to the asynchronous enfacement condition.

**Sense of agency.** The ART-ANOVA results showed a significant main effect of *Synchrony* [$F(1,66) = 21.32$, p < 0.0001, $\eta_p^2 = 0.244$); Fig 3], no main effect of *Gender* (p = 0.47, $\eta_p^2 = 0.0078$), and no significant interaction of *Gender* by *Synchrony* (p = 0.22, $\eta_p^2 = 0.0225$). Thus, similarly to the ownership component, the synchronicity of movements between one' own face and the seen filtered face significantly increased participants' sense of agency.

### How the enfacement illusion affects the perception of the observed face

**Similarity rating.** The ART-ANOVA results showed a significant main effect of *Synchrony* [$F(1,66) = 6.78$, p = 0.0114, $\eta_p^2 = 0.0932$); Fig 4], no main effect of *Gender* (p = 0.57, $\eta_p^2 = 0.005$) and no significant interaction of *Gender* by *Synchrony* (p = 0.055, $\eta_p^2 = 0.0546$). Therefore, participants felt the seen face as more similar to themselves after the synchronous compared to the asynchronous enfacement condition.

**Attractiveness rating.** The parametric ANOVA results showed a main effect of *Gender* [$F(1,66) = 5,89$, p = .018, $\eta_p^2 = 0.08$; Fig 4), a no significant main effect of *Synchrony* (p = 0.410, $\eta_p^2 = 0.0009$)], and no significant interaction *Gender* by *Synchrony* (p = .619, $\eta_p^2 = 0.0038$). In particular, male participants expressed higher attractiveness ratings for their filtered face compared to the female participants regardless of the enfacement illusion condition.

### How the enfacement illusion affects emotional state

The ART-ANOVA results showed no significant main effect of *Gender* [$F(1,66) = 3.48$, p = .067, $\eta_p^2 = 0.049646$)]; no main effect of *Synchrony* (p = .081, $\eta_p^2 = 0.045337$) and no

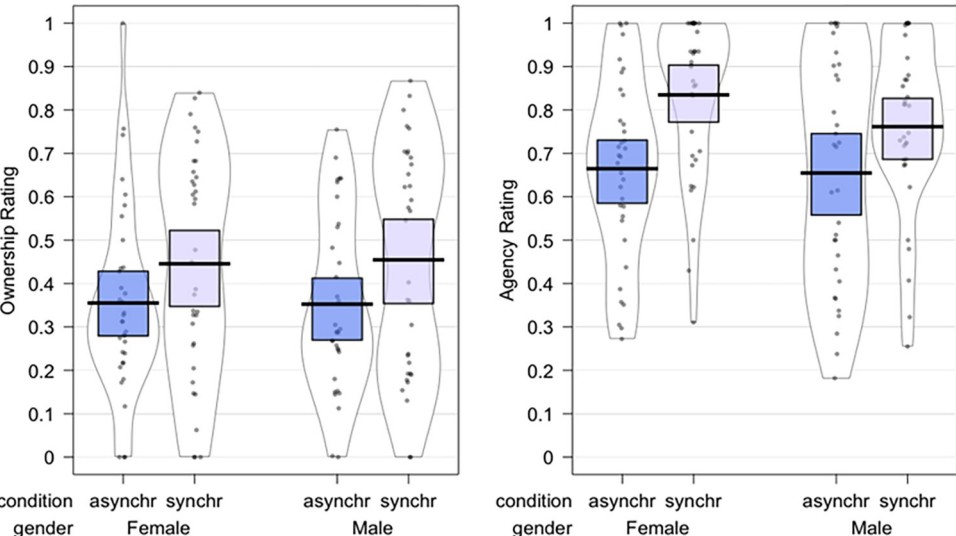

**Fig 3. Plots depicting the raw values, central tendencies, and distribution of the perceived ownership and similarity.**

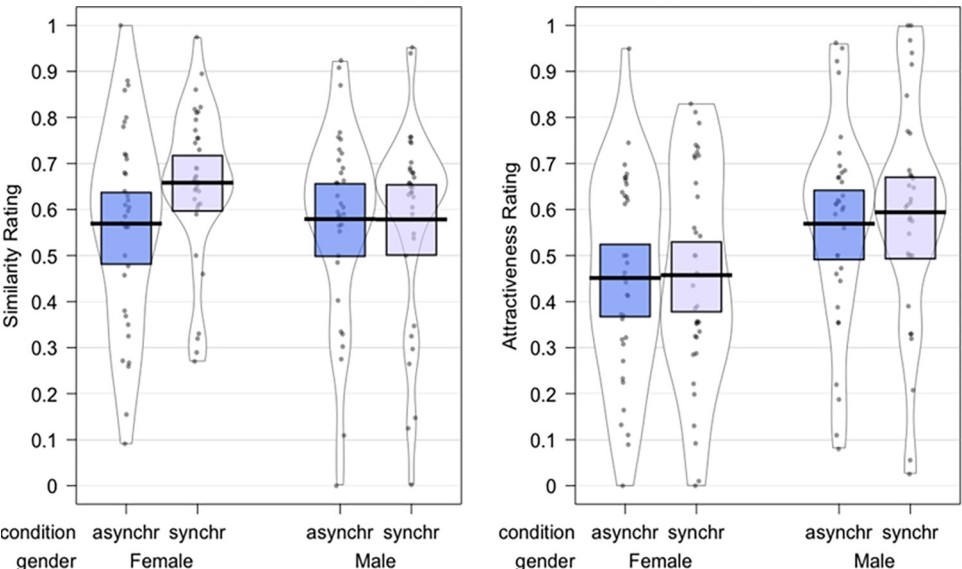

**Fig 4. Plots depicting the raw values, central tendencies, and distribution of the perceived similarity and attractiveness.**

significant interaction of *Gender* by *Synchrony* (p = .161, $\eta_p^2$ = 0.029524). Therefore, exposure to the filtered faces had no effect on participants' emotional state.

## How the enfacement illusion affects implicit and explicit gender identity

**IAT.** The results of the parametric ANOVA showed that the main effect of *Gender* (p = 0.304, $\eta_p^2$ = 0.02) and *Synchrony* (p = 0.304, $\eta_p^2$ = 0.02) and the interaction *Gender by Synchrony* (p = 0.363, $\eta_p^2$ = 0.0003) were all non-significant. A positive D-score reflects a faster reaction towards congruent stimuli, suggesting a stronger implicit association between congruent concepts. Participants showed a relatively low D-score for the synchronous (*Mean* = 0.19, *SD* = 0.42) and the asynchronous (*Mean* = 0.12, *SD* = 0.32) conditions suggesting that in both conditions participants experienced only a weak gender identification (Fig 5).

**BSRI-R.** For the gender congruent items condition, the ART-ANOVA results showed no main effects of *Gender* (p = 0.6285, $\eta_p^2$ = 0.0035), no effect of *Synchrony* (p = 0.2661, $\eta_p^2$ = 0.0187) and no significant interaction *Gender* by *Synchrony* (p = 0.6352, $\eta_p^2$ = 0.0034).

Additional one-sided t-test were performed to determine whether the mean of the asynchronous (*Mean* = 0.018, *SD* = 0.30) and synchronous (*Mean* = -0.10, *SD* = 0.49) conditions (condition values subtracted from their respective baseline values) were significantly different from 0 (Fig 6). Results showed no significant difference between the asynchronous condition and 0 [T(67) = 0.490, p = 0.625] nor between the synchronous condition and 0 [T(67) = 1.63, p = 0.10878].

For the gender incongruent items condition, the parametric ANOVA showed no main effects of *Gender* (p = 0.626, $\eta_p^2$ = 0.003), no main effect of *Synchrony* (p = 0 .476, $\eta_p^2$ = 0.008), and no significant interaction *Gender* by *Synchrony* (p = 0.931, $\eta_p^2$ = 0.0001).

Additional one-sided t-test were performed to determine whether the asynchronous (*Mean* = -0.01, *SD* = 0.29) and synchronous (Mean = 0.05, *SD* = 0.46) conditions (condition values subtracted from their respective baseline values) were significantly different from 0 (Fig 6). Results showed no significant difference between the asynchronous condition and 0 [T(67) = -0.374, p = 0.709] nor from the synchronous condition [T(67) = 0.819, p = 0.415] and 0.

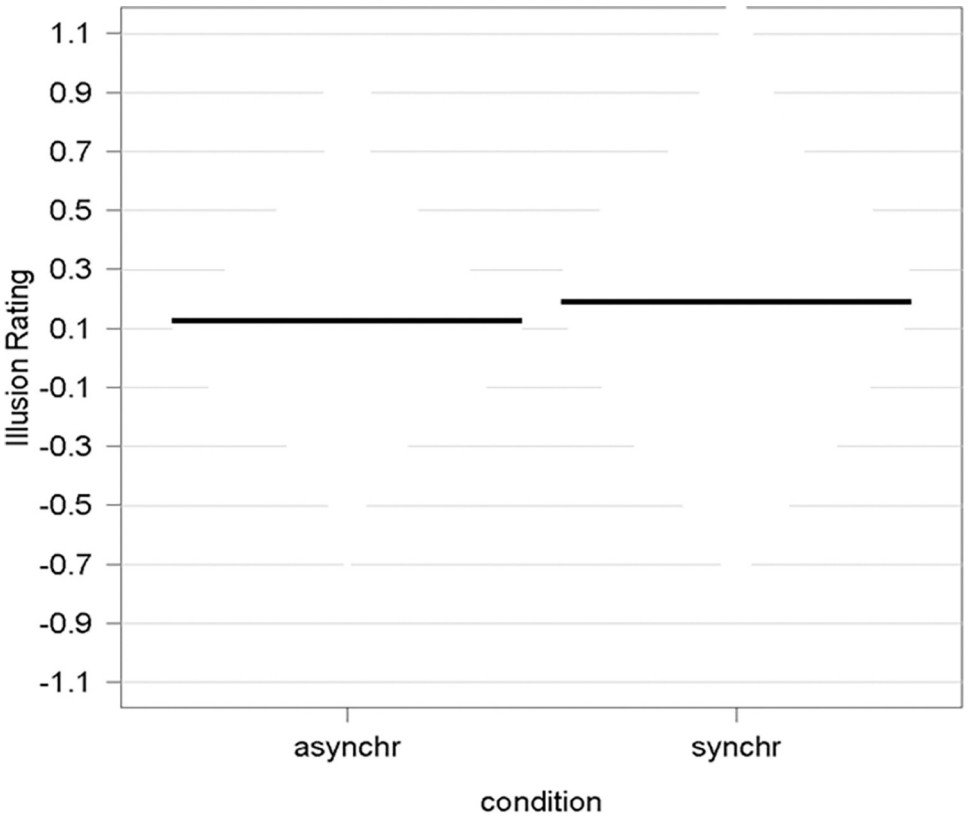

**Fig 5. Plot depicting the D-scores in the asynchronous and synchronous condition.**

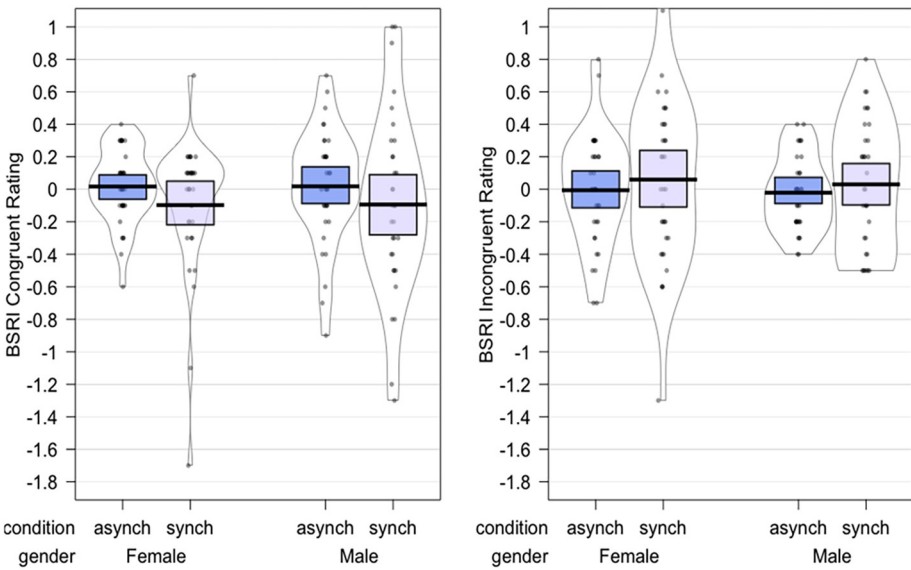

**Fig 6. Plots depicting the difference between the baseline and each condition for each participant's BSRI scores, central tendencies, and distribution for the congruent and incongruent items respectively.**

As absence of evidence does not equate to evidence of absence within the framework of frequentist statistics, we proceeded to enhance our analysis by employing Bayesian multilevel model to investigate the potential support for the null hypothesis (H0). To conduct these analyses, we utilized the R package "brms," which builds upon the rstan package. The application of Bayesian ANOVAs and multilevel models adhered to the methodology outlined in our previous works [36, 37]. In this Bayesian context, posterior probability distributions were generated for the estimated parameters using non-informative priors. The Hamilton Monte Carlo sampling algorithm was employed to derive samples from the posterior distribution of each parameter. To ensure reliable results, we employed four separate Markov chains, each consisting of 1000 warm-up samples and an additional 2000 samples from the posterior distribution. The latter 2000 samples from each chain were retained for subsequent statistical analyses. Convergence of the chains' posterior distributions was verified by computing R-Hat statistics, all of which yielded values below 1.01. This indicated minimal variation between chains in comparison to within-chain variance. For the Bayesian multilevel models, we embraced the maximal random-effects structure that was suitable for our experimental design. This entailed incorporating random effects specific to each participant and condition.

Consistent with the frequentist analyses, the predictors were Gender, Condition and the interaction between the two. Two separate Bayesian models were fitted for BSRI Incongruent and BSI congruent. The 95% Bayesian credible intervals associated with these posterior distributions provided a credible range for the parameter, taking both the data and the model into account. Presence of an effect could be inferred if these credible intervals excluded the value of zero. Consistent with our frequentist counterparts, the Bayesian multilevel model indicated the absence of effects pertaining to Gender, Condition, and the interaction between Gender and Condition. This confirmation of the null hypothesis held true for both BSRI Congruent and BSI Incongruent conditions, as indicated by the crossing of the Credible Intervals over zero.

Finally, in the analysis of the Implicit Association Test, our Bayesian multilevel model, consistent with our frequentist analysis, also found no significant effects for Gender, Condition, or their interaction, as credible intervals span zero.

## Masculinity- femininity VAS

The ART-ANOVA results showed a significant main effect of *Gender* [$F(1,66) = 8.8488$, $p = 0.0041$, $\eta_p^2 = 0.1182$)], no effect of *Synchrony* ($p = 0.8605$, $\eta_p^2 = 0.0005$), and no significant interaction *Gender* by *Synchrony* ($p = 0.1620623$, $\eta_p^2 = 0.0294$). Thus, regardless of the enfacement condition, our male participants reported a higher explicit identification with their self-identified gender, i.e., they felt more masculine, compared to how much the female participants felt feminine. Additional one-sided t-test were performed to determine whether the mean of the asynchronous (*Mean* = -0.27, *SD* = 0.22) and synchronous (Mean = -0.27, *SD* = 0.22) conditions (condition values subtracted from their respective baseline values) were significantly different from 0 (Fig 7). Results showed a significant difference between asynchronous ($T(67) = -10.169$, $p < 0.0001$), and synchronous ($T(67) = -10.219$, $p < 0.0001$) means from 0. Therefore, male and female participants felt less masculine/feminine after observing the filtered faces regardless of synchronicity.

## Discussion

The present study investigated whether experiencing illusionary ownership and agency over a gender-swapped version of one's own face may have an effect on gender identity. Our findings show a successful induction of the enfacement illusion through widely used digital face filters as measured by the questionnaire. These digital filters modulate facial features according to

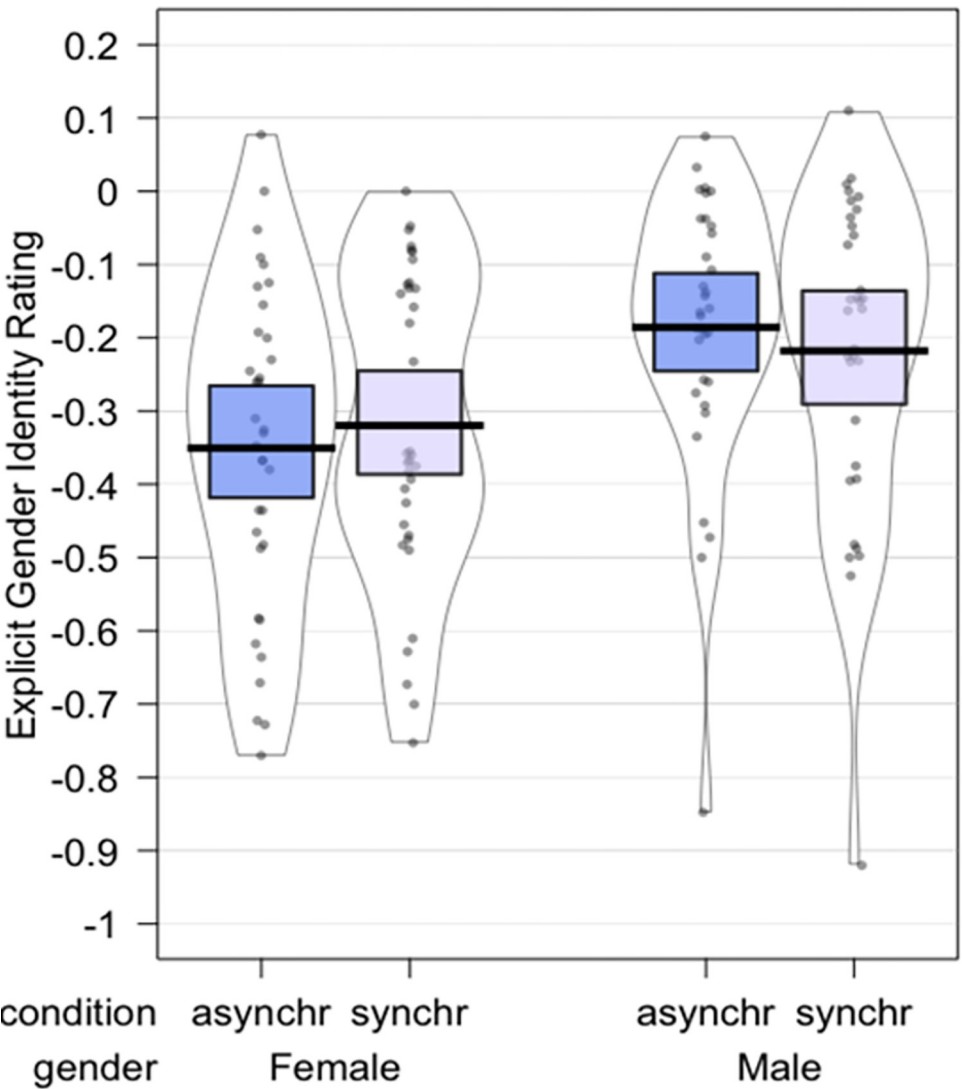

**Fig 7. Plots depicting the difference between baseline and each condition for each participant, the central tendencies, and distribution of the explicit gender identity question.**

stereotypical gender cues in real time. In our study, both male and female participants had a stronger feeling of looking at themselves in the mirror when the filtered face moved synchronously with their own face compared to asynchronously, a very stable and classic finding in the embodiment literature. Furthermore, not only embodiment but also the perceived similarity with the seen filtered face was higher in the synchronous as compared to the asynchronous condition. This effect was previously evidenced in the classical enfacement illusion [38]. These findings are in line with extensive literature on enfacement and with the theoretical principles of embodiment reaffirming that congruent multisensory signals are constitutive to the sense of body [16, 39]. On the contrary, in the present study there is no evidence that such successful induction of the filter-based enfacement illusion would in turn modulate gender identity according to our IAT, BSRI-R scale and explicit gender identity scale collected data. Below, we discuss the peculiarities and relevance of our study, its limitations and possible avenues for future research.

### Enfacement illusion through digital face filters

To achieve the enfacement illusion most studies have used either real people [8, 15, 16, 40, 41], images or videos of real people, self-other face morphs [14] or computer-generated avatars [18]. In contrast to these stimuli and tools developed ad hoc by researchers and accessible only in a laboratory setting, we used widely accessible and popular social media face filters (Snap Inc., Santa Monica, CA). As expected, we show that a short period of active observation of filtered faces moving in synchrony induces subjective effects of the enfacement illusion. This suggests that millions of users might regularly go through short alterations of their bodily self-consciousness. It is important to remind that in our study participants were exposed to the digital face filters for only a brief period of time, whereas the general population's exposure to these filters is much more prolonged and frequent, as the average usage of social media in most western countries is over 3 hours daily [42, 43]. Moreover, on average 6 billion snaps, i.e., photos taken through Snapchat digital lens, are created by the user each day [44]. Thus, it might be that we missed illusory effects regarding gender identity that might occur frequently in daily life as a result of longer and repeated exposures to a stereotypically gender mismatched version of oneself.

Is also worth noting that whereas individuals who participated in our study were all young adults, in the general public social media are frequently used also by a younger adolescent population (97% of adolescents between the ages of 13 and 17 use at least one social media, 67% reported to use Snapchat; [23]), in a development period crucial to gender identification and characterized by the appearance of secondary sexual characteristics [45], where gender identity might be much more amenable to changes derived from exposure to stereotypically gender mismatched version of oneself.

Considering all this, and given the growing accessibility of such devices and platforms, as well as potential behaviours or cognitive alterations that might emerge from their daily use, these initial findings might be of social and clinical relevance [26]. One such alteration is the syndrome known as 'Snapchat dysphoria,' which encompasses body image distortions stemming from the altered digital representation of ourselves. These distortions may perpetuate desirable attributes according to the norms set by digital face filters [25]. As mentioned before, these filters are widely used by communities in their teenage years, a crucial period in forging one's own identity [25, 45]. This is important, as illusory and transient alterations of the bodily-self have already shown to have an effect on both affective, cognitive and social processes as well as behaviour [6, 7, 16, 46]. For instance, a recent study by Burnell and colleagues [47] has found that taking more photos using Snapchat lenses was associated with greater body image concerns. Therefore, it will be important in the future to investigate how the regular usage of these face altering filters might affect key aspects of our self-identity such as our body image and body self-consciousness, especially during a critical phase like adolescence [48].

### No clear changes in gender identity according to our measures

In the present study we based our measures on research that showed a modulation of gender identity co-occurring with changes in the embodiment of a full body [13]. In such a study, participants would embody another real person with a different gender identity from theirs. This was achieved by visually portraying the other person's full body on a head mounted display and synchronously combining visual and tactile signals on the own and the seen body. Similarly to our study, they used an asynchronous stimulation condition for control. While the authors found a relation between self-identifying with the seen body during synchronous stimulation and changes in gender identity, here we found no changes according to very similar

measures. We think that some methodological differences between our and their study may have led to these divergent findings.

On the one hand, we focused on facial instead of body alterations. The face is arguably the most distinct visual characteristic of humans and is a core aspect of self-identity [16]. While some key attributes of faces, such as symmetry along the vertical axis, relatively fixed spatial relationships and social significance are also common to the body [49], others are arguably unique. Not only are body and face processing subject to different developmental periods (e.g., [50, 51]) and neural regions (e.g., [52, 53]) but, more importantly in this setting, one's own body is constantly experienced from an egocentric frame of reference, while the face can only be observed from an allocentric frame of reference and in special conditions (e.g., a reflection in a mirror or screen). Such distinctions between facial and full-body processing–in particular the key role of the own face for self-identity–might be different enough to justify diverging findings between experimental manipulations of the full-body and the seen face. For instance, plenty of experimental evidence has shown that it is possible to experience embodiment illusion over full bodies even strongly different than one's own and this in returns affects perception and attitudes towards the real body [54–56]. This is less clear for the face, and we speculate that could be a protective mechanism preserving one's own self-identity that prevents the enfacement illusion to affect gender's identity, i.e., body and face representation may have a different plasticity due to their different impact on self-identity.

It should further be pointed out that, while highly sophisticated, these digital face filters are generally used for entertainment and in some cases may have generated faces perceived by the participants as odd or not particularly realistic. It is possible that some of our participants may have even experienced an uncanny valley like phenomenon in which a real-life reproduction of a human face induces discomfort and aversion in the observer [57], which is something that we think would be important to control in future studies that intend to employ similar digital filters.

On the other hand, to induce the enfacement illusion we used visuomotor signals while Tacikowski and colleagues [13] relied on visuotactile cues for inducing the embodiment illusion. Visuomotor cues for the enfacement induction have previously been shown to work [17–19]. However, most research on enfacement and its influence on self-perception and identity has been performed using visuotactile signals [58]. While the differential contribution of visuomotor and visuotactile signals to the sense of body is still subject to debate [59–61], this again has primarily been studied in the context of full-body illusions. It might be informative to perform a similar study using popularly used face filters while employing visuotactile cues. We opted for visuomotor cues for two reasons: the first reason is that this is the mode in which these face filters are regularly used in ecological settings, and the second reason is that the visual occlusion caused by tactile stimuli would have interfered with the processing of the face required for the face filters to work properly.

Finally, our results might point to an attenuation of gender identity, even in the asynchronous condition. The (explicit) self-report measure showed an attenuated gender identity after both synchronous and asynchronous stimulation compared to baseline. This might suggest that a more neutral gender identity was induced by both conditions. Unfortunately, for the IAT no baseline measure was taken for experimental length reasons, but the generally low D-scores would be in line with a similar transition. As this is only a speculation so far, future studies would need to address this more specifically. The mentioned change from baseline for both conditions might suggest that temporal synchrony may not be the strongest cue for modulating gender identity in the context of facial cues, but that just perceiving one's own face with enhanced stereotypical gender cues could be sufficient. Recognizing oneself in a photograph does not equate to having a sense of body ownership of one's depiction in that

photograph. The sense of body ownership has unique characteristics such as attributing own-experiential qualities to the self-identified object [62, 63]. Our used filters included participants' own arguably recognizable facial features in both the synchronous and asynchronous conditions which might have impaired the effects of the synchrony modulation on our measures of gender identity. However, many enfacement studies have worked with morphed images containing recognizable features of the participants and still have found alterations in other aspects of cognition related to temporal synchrony (e.g., [14]). In fact, recognition of one's own face has been suggested to be entangled with multisensory integration processes and thus highly malleable [14]. Beyond ownership, agency, and temporal synchrony, recognition might play a role in modulating gender identity through face mirroring; however, our experiment does not allow us to disentangle these aspects. Future studies are encouraged to assess self-recognition in a similar face filter setting. An alternative explanation to the lack of modulation by synchrony could be the relatively high scores in the asynchronous condition, suggesting that a 1s delay might not be optimal for visuomotor facial settings (a 1s delay is used in the visuotactile enfacement literature; see e.g., [16]).

## Limitations and outlook

Among our gender identity measures, we applied an auditory version of the IAT following the work by Tacikowski and colleagues [13]. The reasoning behind this was not only to replicate their setup but more importantly to keep visuomotor congruency between one's own face and the seen face, thus sustaining the enfacement illusion while performing the task. However, during this task, it was qualitatively observed that participants did not move the face much despite being instructed to do so. This may be due to the cognitive resources necessary for attending the IAT task (dual task). In this sense, the differences between enfacement conditions during the IAT might not have been as salient. In principle, we could have tested participants while applying a visuotactile stimulation to sustain the enfacement illusion despite the participant's focus on the task. However, this would have disrupted the currently available face filters from working correctly. In light of these factors, future studies should contemplate administering the IAT after the enfacement conditions and without the vision of the filtered face to mitigate these confounding effects.

Regarding the lack of clear changes in gender identity according to our measures, one potential explanation could be found the in absence of conflict between the observed layered faces and participants' own faces. According to predictive coding theory [64], the presence of congruent spatio-temporal sensory information between one's own face and another person's face induces a conflict, as the observed face does not match the stored representation of the self-face [16]. To solve this conflict, self-representation may change in accordance with the characteristics of the enfaced "other face" or the embodied "other body", as seen in the study by Tacikowski and colleagues [12]. It is possible, then, that in our study, participants did not perceive the filtered face as a conflicting stimulus, and as a result, enfacement did not necessitate significant changes in self-representation.

Another aspect to consider is that we applied a 1s delay for the asynchronous enfacement condition. While this temporal mismatch has been widely used in existing literature (see [16]), to our knowledge no thorough psychometric study has assessed the ideal temporal mismatch for reducing self-identification with one's own seen face for visuotactile ques. This might be a fruitful avenue for future research and could follow existing paradigms working with full body setups (e.g., [59, 63]). It should also be considered that in this study we tested gender as a between-subjects variable, but it might also be important to assess this as a within-subjects variable and allow participants to perceive different gender stereotypical depictions of their own face.

While our use of readily available and popular tools provides important research opportunities, there are also some constraints. The used Snap filters are developed by Snapchat and are not open source, so there is no access to any potential differences in the processing that might be involved in the male versus the female face filters. Furthermore, the company updates the filters automatically and it is necessary to be online in order to use them, therefore it is difficult to know whether certain improvements or changes in the algorithm governing the face filters might have influenced our results. In this sense, potential collaborations with the industry might deem important, allowing researchers to test different aspects of their widely used algorithms. Here we did not investigate prior exposure of our participants to these filters nor their frequency of using this type of technology. This might not only play an important role in individual variability but may also shed light on how the extended use of these tools might influence both the plasticity of the bodily self and its effect on self-identity.

As a cautionary note, we want to state that implicit depiction of gender as a binary that might be associated with this study are confined by the availability of tools and to reduce experimental complexity. The authors do not intend to perpetuate stereotypical depictions of gender as a binary nor untruthful and potentially harmful associations to gender stereotypes. Furthermore, we by no means intend to undermine the long-term struggle of transgender communities to have recognition of their rights. These communities have raised their concerns about digital gender swapping filters (e.g., [22, 65]) and it deems important to honour these voices.

## Supporting information

**S1 File.**
(DOCX)

**S2 File.**
(PDF)

**S3 File.**
(PDF)

## Author Contributions

**Conceptualization:** Bigna Lenggenhager, Marte Roel Lesur.

**Data curation:** Hanna Gohlke, Gianluca Saetta.

**Formal analysis:** Hanna Gohlke, Gianluca Saetta.

**Funding acquisition:** Bigna Lenggenhager.

**Investigation:** Hanna Gohlke.

**Methodology:** Luca Provenzano, Ilaria Bufalari, Bigna Lenggenhager.

**Software:** Luca Provenzano, Marte Roel Lesur.

**Supervision:** Ilaria Bufalari, Bigna Lenggenhager.

**Writing – original draft:** Luca Provenzano, Hanna Gohlke, Gianluca Saetta, Marte Roel Lesur.

**Writing – review & editing:** Luca Provenzano, Hanna Gohlke, Ilaria Bufalari, Bigna Lenggenhager, Marte Roel Lesur.

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
