## [Decision Letter · Decision Letter 0]

9 Aug 2023

PONE-D-23-17502Fluid face but not gender: enfacement illusion through digital face filters does not affect gender identityPLOS ONE

Dear Dr. PROVENZANO,

Thank you for submitting your manuscript to PLOS ONE. After careful consideration, we feel that it has merit but does not fully meet PLOS ONE’s publication criteria as it currently stands. Therefore, we invite you to submit a revised version of the manuscript that addresses the points raised during the review process.

We look forward to receiving your revised manuscript.

Kind regards,

Valentina Bruno

Academic Editor

PLOS ONE

3. We note that Figures 1 and 2 in your submission contain copyrighted images. All PLOS content is published under the Creative Commons Attribution License (CC BY 4.0), which means that the manuscript, images, and Supporting Information files will be freely available online, and any third party is permitted to access, download, copy, distribute, and use these materials in any way, even commercially, with proper attribution. For more information, see our copyright guidelines: http://journals.plos.org/plosone/s/licenses-and-copyright.

a. You may seek permission from the original copyright holder of Figures 1 and 2 to publish the content specifically under the CC BY 4.0 license.

Reviewers' comments:

Reviewer's Responses to Questions

**Comments to the Author**

1. Is the manuscript technically sound, and do the data support the conclusions?

Reviewer #1: Yes

Reviewer #2: Yes

2. Has the statistical analysis been performed appropriately and rigorously? 

Reviewer #1: Yes

Reviewer #2: Yes

3. Have the authors made all data underlying the findings in their manuscript fully available?

Reviewer #1: Yes

Reviewer #2: Yes

4. Is the manuscript presented in an intelligible fashion and written in standard English?

Reviewer #1: Yes

Reviewer #2: Yes

5. Review Comments to the Author

Reviewer #1: The study by Provenzano and colleagues investigated how the enfacement illusion induced with digital face filters affected gender identity. Participants observed their faces with a gender mismatched filter applied; their facial movements were synchronous (experimental) or asynchronous (control) with the movements they observed. Gender identity that was measured during and after the enfacement illusion with a set of implicit and explicit measures was not modulated.

The study addresses an interesting question, its design and analyses are correct, the methods and results are reported in full. I have several minor comments:

Methods:

- Were all participants cisgender?

- Why was the gender identity IAT administered together with the enfacement illusion (participants still saw their filtered face during the IAT), rather than after it, as is commonly done in similar studies with embodiment of differently looking virtual bodies? If it was a replication of the experimental design used in the study by Tacikowski et al. (2020), could it have anyway contributed to the absence of significant effects in the IAT results? For example, a face, especially one considered one’s own, might be a much greater distractor compared to a body. This latter point (attention demands during the IAT+enfacement) is mentioned in the discussion but should it be expanded?

Discussion: paragraph on Enfacement illusion through digital face filters repeats several statements from the introduction almost in the same words and does not refer to any of the results of the study. Would it be better if those arguments were presented in the context of the results?

Reviewer #2: In the current manuscript, Provenzano and colleagues induced an enfacement illusion using a rendered version of the participants' own faces, altered to appear as the opposite gender. Based on a previous study using the full body illusion, authors investigated if the enfacement illusion could modulate not only their physical mental representation but also their gender identity. To this aim, authors compare an illusion synchronous condition, in which the modified face moved synchronously to the participants face, and an asynchronous control condition. Results showed a successful induction of the enfacement illusion in the synchronous condition according to subjective questionnaire ratings, reported agency and perceived similarity. However, authors found no modulation in gender identity, neither in explicit nor in implicit measures.

The manuscript is well-written, the hypotheses are clear, the statistical analyses are sound, and the discussion aligns with the main results. I only have a few minor concerns:

1) The details of the enfacement induction are not entirely clear to me. Did participants follow specific instructions for moving their head, eyes, and mouth? Or did they engage in free, exploratory movements? If the latter, it's possible that such free movements might have contributed to increase the variability between participants in the strength of the illusion. Additionally, could you provide information about the duration of the illusion induction?

2) Another distinction from Tacikowski et al. is that the stimuli used in the enfacement illusion were derived from the participants' own faces. Consequently, "the other" in this study corresponded to a female/male "counterpart" of the participants, rather than an entirely distinct stranger. This aspect might have influenced the outcomes, as individuals may be more inclined to apply gender stereotypes to strangers. As a result, participants may not have merged self-other concepts as effectively, given that the "other" resembled more of a doppelganger than a completely unfamiliar individual onto whom stereotyped gender concepts could be projected.

3) Related to the previous point, were participant aware that they were watching their counterpart in the video?

4) Authors could think to implement some Bayesian analysis to further strength their negative effect on gender identity changes.

6. PLOS authors have the option to publish the peer review history of their article (what does this mean?). If published, this will include your full peer review and any attached files.

Reviewer #1: No

Reviewer #2: **Yes: **Mariano D'Angelo

---

## [Author Response · Author response to Decision Letter 0]

28 Sep 2023

Dear Prof. Valentina Bruno,

We addressed the following points raised in your Decision Letter:

1. We should have formatted the manuscript to meet PLOS ONE's requirement and named the attached files appropriately.

2. We added the information regarding the ethics statment in the manuscript (lines 106-108) and information regarding the written consent obtained from participants, lines 127-128.

3. We attached the consent forms from the copyright holders to publish Figure 1 and Figure 2 (please see files named "CC BY_Content-permission-form_Hanna Gohlke_signed and "CC BY_content-permission-form_Maxwell_Drimmer_signed").

We also added information about copyright usage to both Figure and Figure 2.

4. We include captions for Supporting Information at the end of the manuscript and provided separete files for Supporting Information Figures.

We hope these changes will be found satisfactory,

Best Regards,

Luca Provenzano

---

## [Decision Letter · Decision Letter 1]

20 Nov 2023

Fluid face but not gender: enfacement illusion through digital face filters does not affect gender identity

PONE-D-23-17502R1

Dear Dr. PROVENZANO,

We’re pleased to inform you that your manuscript has been judged scientifically suitable for publication and will be formally accepted for publication once it meets all outstanding technical requirements.

Kind regards,

Valentina Bruno

Academic Editor

PLOS ONE

Additional Editor Comments (optional):

Reviewers' comments:

Reviewer's Responses to Questions

**Comments to the Author**

1. If the authors have adequately addressed your comments raised in a previous round of review and you feel that this manuscript is now acceptable for publication, you may indicate that here to bypass the “Comments to the Author” section, enter your conflict of interest statement in the “Confidential to Editor” section, and submit your "Accept" recommendation.

Reviewer #1: All comments have been addressed

Reviewer #2: All comments have been addressed

2. Is the manuscript technically sound, and do the data support the conclusions?

Reviewer #1: (No Response)

Reviewer #2: Yes

3. Has the statistical analysis been performed appropriately and rigorously? 

Reviewer #1: (No Response)

Reviewer #2: Yes

4. Have the authors made all data underlying the findings in their manuscript fully available?

Reviewer #1: (No Response)

Reviewer #2: Yes

5. Is the manuscript presented in an intelligible fashion and written in standard English?

Reviewer #1: (No Response)

Reviewer #2: Yes

6. Review Comments to the Author

Reviewer #1: (No Response)

Reviewer #2: The authors addressed all my concerns clearly. The statistichal analysis has been perfomed appropiatly and I do not have any further comment

7. PLOS authors have the option to publish the peer review history of their article (what does this mean?). If published, this will include your full peer review and any attached files.

Reviewer #1: No

Reviewer #2: **Yes: **Mariano D'Angelo

---

## [Editor Report · Acceptance letter]

16 Jan 2024

PONE-D-23-17502R1 

PLOS ONE

Dear Dr. Provenzano, 

I'm pleased to inform you that your manuscript has been deemed suitable for publication in PLOS ONE. Congratulations! Your manuscript is now being handed over to our production team.

Kind regards, 

on behalf of

Dr. Valentina Bruno 

Academic Editor

PLOS ONE